Quantification of the influence of drugs on zebrafish larvae swimming kinematics and energetics

Zhao Zhenkai 1
Li Gen 2
http://orcid.org/0000-0001-8512-5299 Xiao Qing 1 qing.xiao@strath.ac.uk
Jiang Hui-Rong 3
Tchivelekete Gabriel Mbuta 4
Shu Xinhua 4
Liu Hao 5
1 Department of Naval Architecture, Ocean, and Marine Engineering, University of Strathclyde , Glasgow , UK
2 Department of Mathematical Science and Advanced Technology, Japan Agency for Marine-Earth Science and Technology (JAMSTEC) , Yokohama-City , Japan
3 Strathclyde Institute of Pharmacy and Biomedical Sciences, University of Strathclyde , Glasgow , UK
4 Department of Biological and Biomedical Sciences, Glasgow Caledonian University , Glasgow , UK
5 Graduate School of Engineering, Chiba University , Chiba , Japan
Gomez Shawn
Electronic publication date: 2020 Jan 8
Publication date: 2020
Volume: 8
Electronic Location ID: e8374
Received 2019 Sep 4; Accepted 2019 Dec 9
Copyright: © 2020 Zhao et al.
Copyright year: 2020
Copyright holder: Zhao et al.
License: This is an open access article distributed under the terms of the Creative Commons Attribution License, which permits unrestricted use, distribution, reproduction and adaptation in any medium and for any purpose provided that it is properly attributed. For attribution, the original author(s), title, publication source (PeerJ) and either DOI or URL of the article must be cited.
License URL: https://creativecommons.org/licenses/by/4.0/

Keywords: Zebrafish, Swimming behaviour, Acetic acid, MBDyn, Fluid-structure interaction, Power, Neuroactive, Drug influence

Funding: Cirrus UK National Tier-2 HPC Service at EPCC funded by the University of Edinburgh and EPSRC EP/P020267/1 Rosetrees Trust PhD studentship from the Angola government This work used the Cirrus UK National Tier-2 HPC Service at EPCC funded by the University of Edinburgh and EPSRC (EP/P020267/1). This work was also partially supported by the Rosetrees Trust. Gabriel Mbuta Tchivelekete is funded by a PhD studentship from the Angola government. The funders had no role in study design, data collection and analysis, decision to publish, or preparation of the manuscript.

==============================
The use of zebrafish larvae has aroused wide interest in the medical field for its potential role in the development of new therapies. The larvae grow extremely quickly and the embryos are nearly transparent which allows easy examination of its internal structures using fluorescent imaging techniques. Medical treatment of zebrafish larvae can directly influence its swimming behaviours. These behaviour changes are related to functional changes of central nervous system and transformations of the zebrafish body such as muscle mechanical power and force variation, which cannot be measured directly by pure experiment observation. To quantify the influence of drugs on zebrafish larvae swimming behaviours and energetics, we have developed a novel methodology to exploit intravital changes based on observed zebrafish locomotion. Specifically, by using an in-house MATLAB code to process the recorded live zebrafish swimming video, the kinematic locomotion equation of a 3D zebrafish larvae was obtained, and a customised Computational Fluid Dynamics tool was used to solve the fluid flow around the fish model which was geometrically the same as experimentally tested zebrafish. The developed methodology was firstly verified against experiment, and further applied to quantify the fish internal body force, torque and power consumption associated with a group of normal zebrafish larvae vs. those immersed in acetic acid and two neuroactive drugs. As indicated by our results, zebrafish larvae immersed in 0.01% acetic acid display approximately 30% higher hydrodynamic power and 10% higher cost of transport than control group. In addition, 500 μM diphenylhydantoin significantly decreases the locomotion activity for approximately 50% lower hydrodynamic power, whereas 100 mg/L yohimbine has not caused any significant influences on 5 dpf zebrafish larvae locomotion. The approach has potential to evaluate the influence of drugs on the aquatic animal’s behaviour changes and thus support the development of new analgesic and neuroactive drugs.

Introduction

In the past decade, the zebrafish has been widely used in medical, biological and genetic research. In its embryonic and larval stage, the zebrafish body is nearly transparent, which conveniently allows the observation of fish organs development. Its quick reproduction speed and cheaper cost, compared to other fish species and mouse, give it a unique and important role in scientific research to resolve a wide range of issues. Among those issues, nociception and nervous system functions are significant and extensively studied. Nociception is a sensory mechanism used to perceive tissue damage (Gregory et al., 2013). Noxious stimuli detected by nociceptors responding to thermal (Malafoglia et al., 2014), electrical (Roques et al., 2010) and chemical (Mettam, McCrohan & Sneddon, 2012) stimulation can cause acute or chronic pain. Zebrafish share similar nociceptive responses to those of human adults (Malafoglia et al., 2013), it is prudent to use the zebrafish larva model to test new analgesic drugs for pain relief. Morphine has been tested on adult zebrafish as an analgesic drug to alleviate the pain caused by acetic acid and shown to have positive effects for pain alleviation. This was achieved via experimental observation and data analysis on fish swimming behaviour changes, such as distance travelled and averaged swimming velocity (Correia et al., 2011; Taylor et al., 2017). Furthermore, more noxious stimuli and drugs have been tested using larvae zebrafish, showing that larvae respond to a noxious challenge in a similar way as adult zebrafish, and the nociceptive response is induced by acetic acid (Lopez-Luna et al., 2017a), which makes it possible to replace protected adult zebrafish with larvae for nociception research.

Administration of neuroactive drugs is an effective method to test animal’s nervous system functions (Irons et al., 2010). As neuroactive drugs acting on different neural pathways could cause different behavioural phenotypes, it is possible to study how the nervous system affects locomotion behaviours by applying different neuroactive drugs (Li et al., 2018). Zebrafish shares similar structure and functions of nervous system compared to mammalian (Anderson & Ingham, 2003; Xi et al., 2010), and have been validated to study neural effects on behavioural manifestation (Sison et al., 2006). A commonly used drug ethanol was studied to determine the acute and chronic effects on zebrafish behaviours and resulted in different types of behaviour alterations such as time spent active, leaping frequency and distance from stimulus (Dlugos & Rabin, 2003; Gerlai et al., 2000; Gerlai, Lee & Blaser, 2006). Other neuroactive drugs such as cocaine and nicotine were also tested on adult zebrafish and attenuation of swimming activity was observed for both of the two drugs (Draland & Dowling, 2001; Levin, Bencan & Cerutti, 2007; López-Patiño et al., 2008). Furthermore, a zebrafish larvae model was studied with the same drugs used in adults and mammalians and showed similar behavioural responses (Irons et al., 2010), suggesting that zebrafish larvae are sensitive to neuroactive drugs.

However, observations and quantifications of only distance travelled and velocity are not effective in ascertaining the influence of drugs on fish swimming behaviour. Fish swimming kinematics are controlled by consecutive contractions of muscles located along each side of the body, and the muscle contractions are directly driven by motoneurons in the spinal cord, which is part of the central nervous system (Ekeberg, Lansner & Grillner, 1995). The bended body controlled by muscle contractions will interact with the surrounding fluid and change its fluid dynamics to power fish swimming (Voesenek, Muijres & Van Leeuwen, 2018). Under this circumstance, to understand internal muscle mechanics, a useful tool to quantify the association between the effect of drugs on zebrafish swimming behaviour and energetics is required. Although many previous experimental studies managed to visualise fish swimming wake patterns via two-dimensional Particle Image Velocimetry flow visualisation technique (Muller, 2004; Muller, Van Den Boogaart & Van Leeuwen, 2008), this is not sufficient to accurately quantify the influences of the drug on fish swimming, which are mainly reflected via muscle mechanical power and force variation. This is a problem that could be potentially solved with a fully coupled fluid structure interaction approach between the fish and the surrounding water.

Inspired by previous work involving Computational Fluid Dynamics (CFD) simulation on fish swimming (Borazjani & Sotiropoulos, 2008, 2009; Carling, Willams & Bowtell, 1998; Lighthill, 1971; Kern & Koumoutsakos, 2006; Li et al., 2012), a novel nociception-related zebrafish larva model combining a biological methodology and a CFD simulation analysis tool to quantify drug influences on zebrafish locomotion has been developed and described in this article. Specifically, we used not only observation of live zebrafish swimming behaviour, but also a CFD simulation tool to quantify a number of important swimming characteristics, including body forces and consumption power, which are hard to acquire with experiments only. In this study, we have studied the influence of particular concentration of diphenylhydantoin (DPH), yohimbine and acetic acid on zebrafish larvae swimming behaviour. The concentration of 500 μM DPH and 100 mg/L yohimbine were selected based on previous research as neuroactive drugs in our study (Li et al., 2015; Liu et al., 2016). The concentration of 0.01% of acetic acid was selected based on Lopez-Luna’s experiment setup (Lopez-Luna et al., 2017b) for a similar pharmacological study including some behavioural studies. In the work of (Steenbergen & Bardine, 2014), they compared the levels of cyclooxygenase-2 (cox-2) in zebrafish larvae and found that activation of nociceptive pathways in a low-concentration acetic acid environment produced behavioural changes that were accompanied by changes in levels of cox-2. As the associated gene is involved in nociceptive processes (Bingham et al., 2006), it seems reasonable to say that the acid-induced behavioural changes can be attributed to nociception. Therefore, it is appropriate to use acetic acid for nociception study.

By comparing the forward swimming speed and hydrodynamic power of wild type zebrafish larvae immersed in water, acetic acid, yohimbine hydrochloride solution and 5, 5-DPH sodium salt solution, we demonstrate that our developed analysis tool is able to quantify some differences, such as fish body internal force and energy consumption, between the group treated with drugs and the control group. As the methodology can, to some extent, quantify the differences of internal muscle mechanics before and after drug treatment, this study has established a foundation for studying the effects of new drugs on zebrafish larvae behaviours.

Materials and Methodology

Ethics

Animal work was carried out in compliance with the Animal Ethics and Welfare Committee, Department of Life Sciences, Glasgow Caledonian University, and UK Home Office under Project License PPL 60/4169.

Experiment setup

The experiment setup (Fig. 1A) and main methodologies were developed in our previous study for the determination of the toxicity of acrylamide on zebrafish locomotion via a colour preference experiment (Jia et al., 2017). In the present study, 85 dpf (days post-fertilization) wildtype zebrafish (Danio rerio) siblings were divided into four groups: the first group (control group) including 20 larvae was immersed in E3 medium (five mM NaCl, 0.17 mM KCl, 0.33 mM CaCl2, 0.22 mM MgSO4, and 0.1% methylene blue); the second group including 20 larvae was immersed in E3 medium with 0.01% acetic acid for 10 min. The third group including 20 larvae was immersed in 500 μM DPH solution (5, 5-DPH sodium salt) and the fourth group including 20 larvae was immersed in 100 mg/L yohimbine solution (yohimbine hydrochloride). The petri dish was illuminated by a light-emitting diode panel, driven by an adjustable DC power supply (CSI5003XE; Circuit Specialists, Tempe, AZ, USA) to provide a continuous and constant light. A high-speed video camera (EoSens CL MC1362; Mikrotron, Unterschleißheim, Germany) was used to record fish swimming behaviour. The frame rate of the camera was set at 500 frames per second during the entire experiment process. As in the subsequent CFD numerical modelling, the selected fish with a tail beating frequency being less than 70 Hz, thus, 7–8 frames within one beat cycle is sufficient to capture the fish tail motion. The water temperature was set at as 27 °C as this is the common temperature widely used in zebrafish experiments. Before the camera started recording, fish in all groups were allowed to swim freely for about 10 min to adapt to the water environment, once recording started, there was no stimulation to force the fish swimming forward. In this study, only quasi-steady cruising swimming regime is investigated excluding the sudden-start process. This is evident from previous research that cruising with cyclically motion is essential for fish larvae to cover the distance for migration and dispersal (Sancho, Ma & Lobel, 1997). In addition, cruising swimming has been studied extensively, which makes it easier to compare with other researchers’ results.

Figure 1 Experimental method used to extract zebrafish motion equations.

(A) Experiment apparatus for zebrafish swimming video recording. High speed camera is used to capture the fish motion in petri dish. (B) Zebrafish image extracted from one frame of the video. (C) Zebrafish outline expressed with white curve and central of mass expressed with green dot. (D) Zebrafish backbone expressed with white line. (E) Equal-distant divisions of the backbone curve, divided with several green dots. (F) Expression of relative angle between two segments. (G) Intesection angle calculation between each two segments along the backbone.

Data processing algorithm

An in-house MATLAB code was developed and used to post-process the recorded videos and extract zebrafish swimming kinematic characteristics, that is motion equations. Figure 1B depicts the key steps for the process. The original image recorded from the camera was converted to a binary image consisting of the sketch of zebrafish larva only with ‘im2bw’ function in MATLAB image processing toolbox. With some adjustments and ‘bwboundaries’ function in MATLAB, a binary image of zebrafish can be extracted, the entire position vector can be obtained for points distributed on fish outline. All images were skeletonised into a single backbone curve using functions ‘bwmorph’ and ‘thin’ operation.

The coordinated pixels on the backbone curve were then divided into equal-distant segments. These segments were simplified as connected straight lines to calculate relative orientation variation with time between two adjacent segments using MATLAB curve fitting toolbox. Physical representation of the intersection angle is shown in Fig. 1C and the mathematical intersection angle is expressed by Fig. 1D, and calculated with Eq. (1), where i denotes the points numbering from one, and θj is the relative angle between each two body segments. As elucidated by Muller (2004), the travelling wave of curvature travels along the fish body at a near constant rate, thus an averaged frequency was selected for the entire relative orientation functions. Equation (2) represents a sample prescribed motion equation for relative angle θj between each two body segments.

(1) θj=arctan⁡(yi+2−yi+1xi+2−xi+1)−arctan⁡(yi+1−yixi+1−xi)

(2) θj=acos(ωt)+bsin(ωt)

CFD solver and motion solver

Zebrafish larva CFD model

Zebrafish larva model used in OpenFOAM (22) (https://www.openfoam.com/) was built with 51 ellipses extracted from the real fish silhouette and controlled by nine deformation equations as shown in Fig. 2A. To simplify the model, the eyes and fin fold are excluded in the CFD fish model. Density of the fish is assumed to be the same as water. The same average body parameters such as body segments mass and length listed in Table 1 is used for all fish in CFD simulation. The flow field was numerically simulated using the open source CFD toolbox OpenFOAM version 3.0.x. The 3-D computational domain is 15 times the fish body length in the longitudinal (x) direction, 10 times of fish body length in transverse (y) direction and four times of fish body length in perpendicular (z) direction. The overall fluid domain is assumed to be at rest initially. In the simulation, the medium is water, therefore, the kinematic viscosity of the fluid υ, which can be expressed as μρ, is 10−6 m2/s. Pressure boundary conditions are taken as zero gradient for all boundaries except the front and back plane, which are set as symmetry; velocity boundary conditions for fish model were taken as moving wall velocity for all body segments and fixed value for the remaining patches. Local mesh around fish model were depicted in Fig. 2B. Mesh around head region before the vertical dash line shown in Fig. 2C is enlarged to be clearer. Considering the constraints in OpenFOAM regarding large mesh deformation to model self-propelled zebrafish swimming, fully unstructured mesh was used to tolerate the internal mesh deformation. For ellipses with high aspect ratio, the mesh is specially refined at the tips to ensure that enough cells are present to precisely capture the vortex at the tips. The Reynolds number is defined as vLυ, v stands for final constant forward swimming velocity, L is the body length of fish larva, and υ represents the kinematic viscosity. For the entire simulation, Reynolds number is set as 300, which stands for intermediate flow regime.

Figure 2 CFD simulation procedure and mesh geometry of the fish.

(A) Flow chart of data transmission between OpenFOAM and MBDyn. Force and displacement vectors are transmitted between OpenFOAM and MBDyn. (B) CAD geometry of fish body. The fish body composes of 51 ellipses with different aspect ratio fish body is divided into nine sections with black hollow circle. (C) Local mesh on fish body in CFD. To accommodate the local mesh rotation and translation, unstructured mesh is built around fish body. (D) Enlarged mesh formation around fish head region.

Table 1 Detailed zebrafish larva body information.

Body section number	Mass (mg)	Length (mm)	
1	0.0385	0.4	
2	0.0553	0.4	
3	0.0425	0.48	
4	0.0308	0.48	
5	0.0212	0.48	
6	0.019	0.48	
7	0.011	0.48	
8	0.009	0.48	
9	0.003	0.32	
Total	0.23	4	

Hydrodynamic solver

To tackle the CFD mesh motion around zebrafish model, a modified displacementSBRStress motion solver is applied in OpenFOAM. PimpleDyMFoam solver is used to solve the transient, incompressible and single-phase Newtonian fluids. PIMPLE algorithm, a combination of SIMPLE and PISO, is used to address velocity–pressure coupling (Liu et al., 2017). Incompressible laminar Navier–Stokes equation was written in Eq. (3) including the conservation equations of mass and momentum. In this equation, U→ represents the fluid velocity, p is the fluid pressure, ρ is the fluid density, and υ is the fluid kinematic viscosity.

(3) ∇⋅(U→)=0∂U→∂ttimeaccumulation+∇⋅(U→U→)convectiontransport=−1ρ∇psourceterm+∇⋅(υ∇U→)diffusiontransport

The time derivatives uses 2nd order implicit discretization scheme, convection term specifies interpolation schemes for velocity as reconCentral, which is different from linear interpolation schemes, it uses extrapolated gradient-based correction from both sides onto the face, using 1/2 weighting to increase stability for large deformation.

In our case, after receiving position and orientation of the body segments from coupled software, OpenFOAM will interpolate the boundary displacement of fish body to the entire domain to calculate the internal mesh motion. Forces and moments are calculated by integrating the pressure and skin-friction forces over the patches, they calculated forces include pressure forces and viscous forces, which are parallel and perpendicular to the target patch.

Multibody dynamic software

The kinematic analysis was based on MBDyn (24), a free general purpose multibody dynamics analysis software developed by the Department of Aerospace Engineering of Polytechnic University of Milan (Politecnico di Milano). It solves initial value problem in the form of Differential-Algebraic Equations (DAE), integrated in time domain using A/L-stable multi-step integration schemes (Li, 2014). Constraints can be added independently in MBDyn, both for rigid and bended body with six degrees of freedom. As our fish model is not a continuous body and is composed of several rigid body segments, it is convenient to use MBDyn to add multiple constraint equations to control the body deformation. By using reference frame, users are able to specify positions, orientations, linear and angular velocities globally and locally. To be specific, we are able to prescribe the relative body deformation between two adjacent body segments and calculate the forces and power of fish body by coupling with OpenFOAM. Dynamics of a set of rigid bodies is written in the form of Newton–Euler equations, constrained by Lagrange’s multipliers. For unconstrained nodes, the equations of motion are expressed as, (4a) M(x)x˙=q

(4b) q˙=f(x,x˙,t)

where x summarises the n coordinates of the system, M(x) is the mass matrix, q summarises the momentum and momenta moments, and f summarizes the generic force including pressure and viscous force. When the system is subjected to kinematic constraints, the constraints are enforced using Lagrange’s multipliers λ, DAE are set as: (5a) M(x)x˙=q

(5b) q˙+ϕTxλ=f(x,x˙,t)

(5c) ϕ(x,t)=0

The DAE are integrated with implicit A/L stable linear multistep integration schemes and a prediction-correction approach is used (Masarati, Morandini & Mantegazza, 2014). In our current case, each body segment was expressed as one node in MBDyn. Except the ground node were set as ‘static’, node, all of the body nodes are dynamic nodes and every two nodes are constrained by a motion equation fitted with MATLAB curve-fitting toolbox.

Coupling strategy

The above two software are coupled using communication primitives provided by MBDyn. To satisfy convergence criteria, a strong coupling, which enables multi-step interactions at each time step, is used between the two software. The schematic diagram for fluid and zebrafish larva motion coupling is shown in Fig. 2D. As indicated in Fig. 2D, kinematic data are transmitted bi-directionally. Inter-process communication is built with Transmission Control Protocol socket. An external force element in MBDyn allows to communicate positions and orientations of a set of nodes, and the corresponding linear and angular velocities with OpenFOAM, the above data can be transmitted either in global frame or in reference frame. On the other side, once kinematic information is received, the forces and moments are calculated based on the changing positions and orientations and transmitted back to MBDyn. Once the convergence criteria are satisfied in OpenFOAM, OpenFOAM stops sending data to MBDyn, and forces and moments in the latest step would be used by MBDyn and keeps iterating process until convergence. The process is repeated until a final convergent solution is reached.

Results

Validations for multi-body coupling

Grid independence and sensitivity test

A grid independence test was carried out on a self-propelled 0.01% acetic acid treated zebrafish model with fully prescribed deformation with two mesh sizes, medium and fine mesh. In addition, the time step influence is also tested, for example Case 1: Medium mesh (M) with 83,200 total cells and time step of T/650, Case 2: Medium mesh and smaller time step (MS) of 83,200 cells and time step of T/1300, and Case 3: Fine mesh (F) of 166,400 cells and time step of T/650. The size of the fluid domain remains same for three cases.

To exclude mesh resolution and time step size influences on simulation results, we have compared the forward velocity and total force of zebrafish larvae treated with 0.01% acetic acid and the results are shown in Figs. 3A and 3B. Computational results for three cases including kinematic performance and fluid domain calculation shows close results. To save computational time and keep accuracy, we use the mesh formation and time step size the same as Case 1.

Figure 3 Grid independence test with acetic acid treated zebrafish larva sensitivity study.

(A) Forward velocity for three levels of grid. (B) Total force in the moving direction for three levels of grid. (C) Forward velocity for three numbers of segmentations. (D) Total hydrodynamic force for three numbers of segmentations.

In addition, we have also tested the sensitivity of the results to body segments of zebrafish larvae model with CFD simulation. Considering the accuracy and efficiency for body segmentation, we have divided the body trunk into 5, 10 and 15 segments, respectively. By capturing the body deformation and simulate the forward motion with CFD toolbox, we compared the forward velocity and total hydrodynamic force in Figs. 3C and 3D, indicating that the simulation results are not sensitive to the number of body segments.

Multi-body dynamics validation

To validate our numerical coupling methodology on multi-body structure simulation, a comparison based on jellyfish-inspired swimming provided by Wilson was performed (Wilson & Eldredge, 2011). Figure 4A depicts the jellyfish shape, points on the figure indicate how the main body is separated. Figures 4B and 4C illustrate the mathematical model we have created and our CFD model. The prescribed angle functions are represented by θ1, θ2 and θ3. Detailed functions are expressed with Eq. (6). By specifying relative orientation equations between every two sections, the jellyfish model can move upwards with alternant contraction and refilling. Different Reynolds numbers were tested, which is defined by the maximum jellyfish diameter and undulation period, that is D2maxTυ, Dmax is the maximum diameter of the jellyfish model, T is the undulation period, and υ denotes kinematic viscosity. Here we selected Re equivalent to 140 and 70. Figures 5A and 5B depict the kinematic performance of the multi-body structure with fully prescribed motion, while Fig. 5C illustrates the power required for the model to move cyclically and Fig. 5D shows the vorticity comparison at Re = 140. All of our results are in close consistent with Wilson & Eldredge’s results, the small discrepancies can be caused by the different numerical methods used.

(6) θ1=−0.1472cos⁡(6.538t)−0.3247sin⁡(6.538t)+0.7551θ2=−02366cos⁡(6.456t)+0.1645sin⁡(6.456t)+0.3472θ3=−0.08218cos⁡(6.427t)+0.04095sin⁡(6.427t)+0.4511

Figure 4 Jellyfish model.

(A) Real jellyfish shape with dividing points. (B) Mathematical jellyfish model. θ1, θ2 and θ3 are intersection angles between each two segments. As the structure is symmetric to Y axis, values of intersection angles on the other side is equivalent but with reversed sign. Dmax represents maximum diameter of jellyfish model. (C) Local unstructured mesh around CFD jellyfish model.

Figure 5 Validation results for periodic jellyfish movements.

(A) Longitudinal centroid position for Re = 140. (B) Longitudinal centroid velocity for Re = 140. (C) Required input power for Re = 70. (D) Our CFD simulation result of vorticity at Re = 14. (E) Wilson & Eldredge’s (2011) result of vorticity at Re = 14.

Validations on numerical and experimental methodologies

The first objective of our study was to verify the correctness/accuracy and consistence of proposed experiment and CFD method in this work. To achieve this goal, we used the time-varied angle data, extracted via post-processing of experiment video records as input data to CFD fish body discrete elements, the forward motion of fish is the solution by solving a coupled flow solver and multi-body dynamic method as described in “CFD Solver and Motion Solver”. The first and last fish body segment motion is not prescribed whereas they are our numerical solutions, which represent the fish head and tail locomotion. With simulation results, we first compared our yawing head angle and tail beat angle with data collected from observed zebrafish swimming. Figures 6A and 6B show an average head and tail angle for 10 individual zebrafish larvae. As seen from the figures, both the head and tail-beat angles were calculated based on the prescribed constrained deformation equations of zebrafish-matched experimentally observed data. Slight differences can be caused by subtle insufficient accuracy of the fish model capturing toolbox, which might lead to slight inconsistency in certain captured points of head/tail angle. The forward swimming velocity shown in Fig. 6C reaches about 95% of the experimentally measured swimming speed, approximately 19.25 body lengths per second. The small gap exists as the real fish may not bend the whole body as symmetrically as our model, which is fully controlled by prescribed sinusoidal functions. The subtle modelling errors may additionally lead to differences in swimming speed. We have also calculated the Strouhal number (St) for simulated CFD results. The St is defined as AfU, where A is the tail beat peak-to-peak amplitude, f is the tail beat frequency and U is the averaged swimming velocity. In our simulation, the St is approximately 0.8. A statistical data about forward velocity for 10 zebrafish larvae is shown in Fig. 6D (to save space of the image, only ten results are shown).

Figure 6 Comparison of simulated zebrafish data in global frame with experimentally observed zebrafish swimming.

(A) Head angle. (B) Tail angle. (C) Forward swimming speed. (D) Forward swimming speed for 10 fish.

Figure 7 depicts the vorticity iso-surfaces formed based on Q Criterion behind a swimming normal zebrafish larva at different instants in time within one time period and the dorsal view for vorticity iso-surfaces. Here, Q can describe the wake topology and defines vortices as positive second invariant of velocity gradient in region where vorticity magnitude is greater than strain-rate magnitude (Kolář, 2007). As seen from the most left and right column of Fig. 7, flow patterns behind the fish are represented by detached vortices and shown as translucent green fragments. Vortices starts to form in the vicinity of head, transmits downstream to tail and detaches at the tail, which are consistent with the fish tail motion; when the lateral displacement of the tail reaches the highest amplitude, vortices starts to shed at the tail tip, the already formed vorticity in the wake are mixed with the newly formed vorticity at tail tip. The most right column also displays a 3-D view of the vortex rings generated behind the fish to understand formation of flow patterns better. To validate the numerical methodology, Fig. 7 also compares the body curvatures of CFD model and the real fish in the recorded experiment video. As can be seen, two sets of results match very well in terms of body shape at all specific time within a period, indicating that our CFD model is able to imitate the self-propelled swimming of zebrafish larva and its interactions with surrounding fluid.

Figure 7 Vortex rings behind zebrafish larva for Q = 0.5 at different time step within one period of time and the corresponding video record for the experiment.

x-y plane vorticity is a 2-D view of the 3-D vortices which can compare the body curvature with experiment results easier. From (A)–(D), (E)–(H) and (I)–(L), time steps are 0, T/3, 2T/3 and T for each column. T represents one period of time.

Power distribution along fish body and an initial approximation of the cost of transport

As the movement of each two neighbouring body segments is constrained with a prescribed deformation equation except for fish head and tail, mechanical power distribution along the fish body can be approximated by power generated by each body segment. The mechanical power generated from fish muscle includes the translational power due to linear motion and the rotational power due to body rotation. As the fish is moving cyclically, all the other terms are cancelled out except for the rotational power. Therefore, the mechanical power is estimated with the cross product of torque and angular velocity shown as Eq. (7a), and the total power transmitted into the water is calculated with Eq. (7b).

(7a) PM=∑iMi⋅ωi

(7b) PH=∑j−Fj⋅Vj

(7c) PM¯=PH¯

In the above equation, PM is the mechanical power of fish muscle, and PH represents hydrodynamic power generated by interactions with surrounding fluid. Mi is the internal torque for the ith joint calculated by MBDyn in the global frame, ωi represents the angular velocity for the ith joint. Fj is the hydrodynamic force acting on the jth body and Vj represents the jth body velocity.

During muscle contraction, fish body bends, energy is generated and transmitted into the water, and the bended body interacts with the surrounding fluid, a thrust force generates and pushes the fish moving forward. Within this process, Approximately 20% of the energy (depending on fish species, the percentage can float dramatically) (Zhang, Yu & Tong, 2014) generated by muscle contraction is consumed due to the viscous dissipation of fish body tissues, and the remaining energy is transmitted into the water. In our model, viscous dissipations of fish body tissues are neglected to simplify the simulation. Therefore, as the variation of kinetic energy is zero during cyclic swimming, the mechanical power is fully transformed into the hydrodynamic power, which means the absolute value hydrodynamic power equals the mechanical power, and we only need to show hydrodynamic power for 20 zebrafish larvae as depicted in Fig. 8B. The calculated hydrodynamic power has been changed into absolute values as the sign is different from mechanical power. To further understand the different power generated at different locations along the fish body, we have compared the time-history of forces and velocity in Fig. 8 at three typical points shown in Fig. 8A, representing head region, body region and tail region. The trajectory is not completely parallel to X axis in global frame, force and velocity shown in Figs. 8C and 8D are pointing towards the real moving direction of zebrafish. An approximation of hydrodynamic power distribution on each body section is also depicted. As shown in Fig. 8E, the averaged hydrodynamic power for 20 fish larvae shows a significant higher value starting from approximately 75% of body length. According to motion equations, this region has the largest motion amplitude along the body in global frame, resulting in larger fluid force, thus more hydrodynamic power. In Fig. 8F, the mechanical power generated along the body shows an increase towards the tail and a steep decrease at the tail.

Figure 8 Power distribution at three typical points and along fish body.

(A) Real zebrafish picture. (B) Hydrodynamic power for ten sample five dpf zebrafish larvae. (C) Hydrodynamic force at three points. (D) Velocity at three points. (E) Averaged hydrodynamic power for 20 fish larvae distribution of each body section along fish body. (F) Averaged mechanical power distribution of each joint along fish body for 20 fish larvae.

Detailed mechanical and hydrodynamic power and an approximation of cost of transport are summarised in Table 2. The cost of transport is defined as energy spent to travel unit distance per unit mass, which is expressed as PmU, Pm is the power per unit mass, and U is the forward velocity. Limited by the size of the table, only 10 fish data was selected and displayed in the table.

Table 2 Detailed mechanical and hydrodynamic power values for 10 normal zebrafish larva numbered from fish1 to fish10.

		S1	S2	S3	S4	S5	S6	S7	S8	S9	Cost of transport μJ/m	
fish1	Hyd	0.16	0.04	0.05	0.06	0.09	0.14	0.08	0.26	0.55	79.97	
	Mec	0.00	0.03	0.04	0.06	0.21	0.32	0.38	0.31	0.07	
fish2	Hyd	0.17	0.04	0.03	0.05	0.08	0.12	0.08	0.26	0.51	78.78	
	Mec	0.00	0.03	0.04	0.06	0.20	0.31	0.36	0.30	0.07	
fish3	Hyd	0.19	0.04	0.06	0.07	0.11	0.14	0.09	0.26	0.56	84.33	
	Mec	0.00	0.03	0.05	0.09	0.21	0.32	0.41	0.33	0.08	
fish4	Hyd	0.23	0.04	0.05	0.06	0.12	0.15	0.10	0.08	0.63	90.57	
	Mec	0.00	0.04	0.06	0.12	0.22	0.37	0.49	0.34	0.08	
fish5	Hyd	0.18	0.04	0.05	0.07	0.12	0.16	0.09	0.28	0.55	80.64	
	Mec	0.00	0.04	0.06	0.09	0.21	0.31	0.42	0.33	0.08	
fish6	Hyd	0.19	0.04	0.05	0.06	0.08	0.14	0.09	0.28	0.54	77.61	
	Mec	0.00	0.03	0.05	0.06	0.20	0.33	0.40	0.31	0.08	
fish7	Hyd	0.18	0.04	0.06	0.06	0.10	0.14	0.09	0.28	0.59	85.56	
	Mec	0.00	0.04	0.06	0.09	0.22	0.31	0.42	0.33	0.10	
fish8	Hyd	0.18	0.04	0.05	0.06	0.10	0.13	0.09	0.27	0.55	79.66	
	Mec	0.00	0.04	0.04	0.07	0.20	0.32	0.41	0.31	0.08	
fish9	Hyd	0.18	0.04	0.05	0.06	0.10	0.13	0.09	0.27	0.55	79.18	
	Mec	0.00	0.03	0.05	0.06	0.21	0.32	0.42	0.30	0.08	
fish10	Hyd	0.18	0.04	0.05	0.06	0.10	0.12	0.09	0.28	0.61	81.01	
	Mec	0.00	0.04	0.06	0.09	0.21	0.31	0.41	0.33	0.09	
Note:

S1–S9 represents the nine body sections. Hyd, represents hydrodynamic power; Mec, represents mechanical power units of both are uW.

Comparison of kinematics and energetics between normal group and drug treated groups

Swimming performance of zebrafish larvae might be affected by drugs with different concentrations. According to Lopez-Luna’s research (Lopez-Luna et al., 2017b), zebrafish larvae exposed to 0.01% acetic acid displayed more active responses than the normal zebrafish, and these active behaviours were sustained for longer. Liu et al. (2016) has tested influences of (DPH) on 5 dpf zebrafish larvae locomotion at different concentrations and found that exposure to higher concentrations of DPH under light condition leads to decreased locomotor activities. Their team also tested the effect of 100 mg/L yohimbine on 5 dpf zebrafish larvae, but no obvious effect has been found on locomotor activity under light condition (Li et al., 2015). By simulating the swimming behaviours of zebrafish larvae under these circumstance, we examined the forward swimming velocity and cost of transport differences after exposing to 0.01% acetic acid, 500 μM DPH, and 100 mg/L yohimbine. Figures 9A and 9D depict the averaged head and tail comparison for three drugs and control group zebrafish larvae. As the different frequencies and initial angles were used, phase differences existed among those groups in head and tail angle respectively. From the figure, there are no significant differences for the amplitude of head angle, whereas the maximum tail beat angle for zebrafish larva immersed in the 0.01% acetic acid showed a larger value compared with the rest groups. Figure 9B illustrates the forward swimming speed comparison between control group zebrafish and drug treated groups. The forward velocity for acetic acid treated zebrafish is dramatically higher than other groups’ zebrafish, but there is no significant differences between control group and yohimbine treated group. For zebrafish treated with 500 μM DPH solution, there is a significant decrease of velocity compared with other groups. These results indicate that the forward swimming velocity might be influenced by the body undulation frequency and tail beat amplitude. With larger frequency and tail beat amplitude being represented by maximum tail beat angle, the forward swimming speed tended to increase.

Figure 9 Comparisons between control group, 0.01% acetic acid treated group, 500 μM DPH treated group, and 100 mg/L yohimbine treated group.

(A) Head angle. (B) Forward velocity. (C) Hydrodynamic power. (D) Tail angle. (E) Forward velocity comparison for control group and drug treated groups with two-tailed t-test for twenty fish in all, shown in mean (SD); ****P < 0.0001. (F) Total hydrodynamic power generated by surrounding water with two-tailed t-test for control group and drug treated groups, shown in mean (SD); ****P < 0.0001.

As the tail beat frequency and amplitude are both increased, energy generated by fish body will increase as well, which might give rise to lower efficiency as the side oscillations consumes more energy without contributions to thrust. Figure 9C displays comparisons of the hydrodynamic power PH, cost of transport were calculated for all cases, resulting in 81.73 μJ/m·kg, 96.24 μJ/m·kg, 65.44 μJ/m·kg and 82.32 μJ/m·kg for control, acetic acid treated, DPH treated, and yohimbine treated group, respectively. These values are similar to those reported by Li et al. (2016) (from 105 μJ/m·kg to 50 μJ/m·kg) in on larval zebrafish. It appears that the DPH treated group performed with greater efficiency than the other groups as it has the lowest tail beat amplitude and frequency. Figure 9E depicts the forward velocity for all groups. Based on the assumption of no differences among those groups, we have calculated the P value, which is usually evaluated in statistical hypothesis testing to determine the reliability of the results. The P value is smaller than 0.0001 for control group comparison with DPH treated group and acetic acid treated group, which indicates that significant increment of forward velocity can be seen after 0.01% acetic acid treatment, and a decrease of velocity happens after treated with 500 μM DPH solution. There is no obvious differences between control group and yohimbine treated group, indicating that 100 mg/L concentration of yohimbine will not influence the locomotion behaviour dramatically of 5 dpf zebrafish larvae. Figure 9F displays similar comparison as Fig. 9E, but with the parameter changed to hydrodynamic power. The resulting P value is smaller than 0.0001 for control group comparison with DPH treated group and acetic acid treated group, leading to the conclusion that averaged hydrodynamic power of acetic acid treated zebrafish larvae is higher than the control group specimens, whereas the zebrafish larvae treated with DPH solution resulted in lower averaged hydrodynamic power. There is no obvious difference between control group and yohimbine treated group as well, which is consistent with results shown in Fig. 9E. Figure 10 compares the vorticity on x-y plane of the 3-D fish model. By comparing them in one period of time, it can be seen that the vortex detached from the tail tip is faster for acid treated group compared with control group fish. The earlier detached vortex has been labelled with black circles in the right column, indicating the larger distance travelled within one period of time, that is higher velocity. For DPH treated group, within same one period of time, the evolution of vorticity is much shorter than other groups, approximately half of other groups, suggesting that only half of the distance travelled by DPH treated group compared with control group. For yohimbine treated group, vorticity patterns are similar to those in control group. All of the vorticity results for drug treated groups are in consistent with velocity comparisons depicted in Fig. 9.

Figure 10 Vorticity comparisons in x-y plane between control group zebrafish and zebrafish exposed to drug treated groups within one period of time.

For (A)–(D), (E)–(H), (I)–(L) and (M)–(P), time steps are 0, T/3, 2T/3 and T for each column. T represents one period of time.

Discussion

A novel methodology connecting biological experiments and CFD simulation to explore the relationship between zebrafish larvae fish swimming behaviour and intravital body force/torque changes is proposed and tested in this study. By using the observed zebrafish locomotion, we extracted the kinematic swimming equations and entered them into our numerical modelling tool to achieve a fluid-body interaction numerical simulation. Although the estimated final cyclic averaged swimming speed of zebrafish larvae via CFD is slightly lower than the experimentally observed results, overall agreement between experiment and CFD is acceptable. Compared with previous studies on zebrafish larvae (Li et al., 2012, 2016), the resultant swimming motion and energetics are within a reasonable range. The estimated St is around 0.8, which is much higher than optimal streamlined fish swimming value of 0.5, the probable reason might be due to that the zebrafish larvae swims in intermediate flow regime where viscous force dominant, Therefore, overcoming such viscous effect requires more thrust and thus higher St is needed (Voesenek, Muijres & Van Leeuwen, 2018). As shown in Fig. 7, though we can conclude some differences from the stage of vortex detach by comparing the vorticity in the wake that can be acquired from experimental observations, distinctions between control and acetic treated group are still not clear, which might require quantified data from CFD results described in the previous paragraphs to further identify the differences, indicating the potentials of CFD simulation in the comparisons of nociceptive related studies.

We have also calculated the hydrodynamic power distribution along the body as shown in Fig. 5E. Based on our results, the hydrodynamic power generation shows an increase starting from the centre of mass and a steep increase in the rear region. Ideally, the consumption of muscle power requires a study of muscle strain and electromyography patterns for muscle function at specific positions along the body; however, the extremely small body size of larval fish makes it impossible to place receivers on the body. Constraints added in our fish model provide energy to move forward from static state, which perform as muscle fibre in real fish to provide mechanical power. Mechanical power distribution has been examined in Fig. 5F, showing a steep increase towards the tail from the middle region and then a steep decrease in the tail region. This might suggest that the main power generated by muscle to support steady forward swimming exists in the entire body. The conclusion seems to be inconsistent with the previous viewpoint that most power is generated in the anterior region, while the posterior region performs like a transmitter. However, based on the equations set up in our model, the anterior region equations have smaller curvature, implying that the simulated muscle in this region has smaller strain when it is contracted, thus less positive work is done. Moreover, during steady swimming state, red muscle dominates the swimming motion; if the main muscle power is generated in the anterior part, loss of energy in the form of heat occurs in the process of force transmission towards the tail, which might increase the burden of red muscle as it powers the entire steady swimming process (Rome, Swank & Corda, 1993). Given that muscle functions vary among different species, our results need to be further tested with the help of biological analysis.

Acetic acid treated zebrafish might accelerate quicker, that is higher frequency, to reach the maximum speed as it attempts to escape the acid environment (Lopez-Luna et al., 2017a). Our results show that the acid treated group achieved higher tail beat frequency and swimming speed than control group. The increment of speed in the intermediate flow regime (10 < Re < 103) increased the energy dissipation, resulting in higher cost of transport, which agrees with previous study by Li et al. (2012).

Two neuroactive drugs used in this article (DPH) and yohimbine are both sensitive to zebrafish larvae, for different days of post fertilisation, behavioural changes are different (Irons et al., 2010). Besides, different concentrations and lighting conditions can cause reversed results for same drug. For example, for 5 dpf zebrafish larvae, 10 mg/L yohimbine will increase the locomotion activity of fish larvae, whereas 200 mg/L yohimbine will decrease the activity. In our study, 5 dpf zebrafish larvae applied with 500 μM DPH and 100 mg/L yohimbine lead to similar results compared with previous biological observations (Li et al., 2015; Liu et al., 2016), indicating that our method has the ability to replicate neuroactive drug influences on zebrafish larvae locomotion behaviours. The effect of exposure to acid on zebrafish swimming behaviour has been studied for different substances including acetic acid and citric acid and at different zebrafish developmental stages (Lopez-Luna et al., 2017a; Nordgreen et al., 2014). However, these studies were limited to the nociceptive responses of zebrafish larvae on stress, fear or anxiety, that is environment influences. Furthermore, the observed data mainly focused on the total distance fish travelled in a period of time or the time spent in active status (Lopez-Luna et al., 2017a, 2017b). To some extent, our developed tool can mimic the mutual interactions of real fish with the surrounding fluid and thus allows investigation of the relationship between fish body mechanical force and torque and its swimming behaviours. Using this approach, our future work would focus on evaluating potential analgesic drugs for pain relief and neuroactive drug effects on fish behaviours, which might help to understand functions of nervous system.

A possible factor that might influence the accuracy of our results is the fish body stiffness, which has not been taken into account in the present research but has been studied by other groups (McHenry & Van Netten, 2007). Studied the flexural stiffness of superficial neuromasts as it is correlated with the detection of surrounding fluid. Zhang, Yu & Tong (2014) provided a prediction of fish body’s visco-elastic properties and related muscle mechanical behaviour in vivo based on a continuous beam model (Zhang, Yu & Tong, 2014). Real fish are able to adjust their body stiffness at specific positions in order to optimise their swimming performance such as the maximum forward speed and minimum energy cost (Tytell et al., 2016). However, the distribution of visco-elastic properties, that is stiffness and damping coefficients along the fish body, are difficult to measure precisely, thus the mutual contributions from visco-elastic properties to the optimised swimming performance cannot be determined individually. Moreover, it is technically difficult to observe subtle body curvature changes. Different fish species may have different stiffness and damping characteristics for different purposes, such as for acceleration/deceleration or cruising swimming (Tytell, Hsu & Williams, 2010). Considering the importance of body stiffness for a better understanding of muscle functions in controlling fish swimming, we intend to focus in our future research on the visco-elastic properties at some predicted positions with the help of muscle dissection. To be specific, muscle related adverse medical treatment may have effects on muscle tissues such as shortened or dissolved local muscle fibres (Lin, 2012). By applying predicted stiffness and damping coefficients and comparing these with the live fish tissue properties at those locations, it might be possible to account for the influences on altered swimming behaviours.

Conclusions

In this article, a novel method has been introduced to quantify the influence of drugs on zebrafish locomotion kinematics and energetics. Experimental results have compared with CFD simulated results on tail beat angle and forwards velocity and showed consistent values. Three types of drugs with positive effect, negative effect and no effect on zebrafish locomotion activity validated by previous researches have been applied to study applications of our methodology. Reasonable comparison results have been supplied, 0.01% acetic acid has a positive influence on 5 dpf zebrafish locomotion, 500 μM DPH solution has a negative effect on zebrafish locomotion, and the 100 mg/mL yohimbine will not influence swimming behaviours of 5 dpf zebrafish larvae significantly. We have also provide results related to internal muscle mechanics, including power distribution along the body and hydrodynamic power comparison, providing insights into internal muscle influences on fish swimming.

There are still some questions to be solved related to this direction, particularly with respect to how the change of internal muscle could influence swimming behaviours of zebrafish larvae, and how the passive control of fish muscle contributes to swimming efficiency. Evaluation of the analgesic and neuroactive drugs on fish behaviours is also an ongoing effort.

Supplemental Information

Supplemental Information 1 MatLab code for image processing.

This code provides a method to process a selected time range of the recorded video for zebrafish larvae and get its deformation equation

Click here for additional data file.

We thank the support from Dr. Gen Li and Professor Hao Liu on reviewing the manuscript and apparatus provided by Dr. Xinhua Shu for zebrafish experiment.

Additional Information and Declarations

Competing Interests

Author Contributions

Animal Ethics

Data Availability

The authors declare that they have no competing interests.

Zhenkai Zhao conceived and designed the experiments, performed the experiments, analysed the data, prepared figures and/or tables, authored or reviewed drafts of the paper, and approved the final draft.

Gen Li conceived and designed the experiments, authored or reviewed drafts of the paper, and approved the final draft.

Qing Xiao conceived and designed the experiments, authored or reviewed drafts of the paper, and approved the final draft.

Hui-Rong Jiang conceived and designed the experiments, authored or reviewed drafts of the paper, and approved the final draft.

Gabriel Mbuta Tchivelekete performed the experiments, prepared figures and/or tables, and approved the final draft.

Xinhua Shu conceived and designed the experiments, authored or reviewed drafts of the paper, and approved the final draft.

Hao Liu conceived and designed the experiments, authored or reviewed drafts of the paper, and approved the final draft.

The following information was supplied relating to ethical approvals (i.e., approving body and any reference numbers):

Animal work was carried out in compliance with the Animal Ethics and Welfare Committee, Department of Life Sciences, Glasgow Caledonian University, and UK Home Office under Project License: PPL 60/4169.

The following information was supplied regarding data availability:

The raw data is available at Zenodo: Zhenkai Zhao (2019). Data set for control and drug treated group zebrafish larvae. Zenodo. DOI 10.5281/zenodo.3540803.

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
