# Peer review of "Quantification of the influence of drugs on zebrafish larvae swimming kinematics and energetics"

_PeerJ, doi:10.7717/peerj.8374_

## Round 0.1 · original submission · Minor Revisions

Please address the reviewer concerns. In particular, Reviewer 2 points out a number of items that if addressed, would significantly improve the clarity of the manuscript.

Reviewer 1 ·

Basic reporting

The write-up is organized and clear. There were two typos in equations that I have mentioned below.

Experimental design

Methods are well-described and adequately validated.

Validity of the findings

conclusions are fine. Just the statements regarding the need for CFD needs to be softened as I mention below because it was not directly shown in this paper.

Additional comments

For transparency, I had reviewed this manuscript for another journal before. This paper is much clearer (except a few equations as pointed below) and the numerical methods are well-described. The sensitivity and validation studies are adequate. The results are presented clearly. My only concern is that the statements regarding the need for CFD to measure muscle power for drug testing need to be softened because it was not directly shown in the paper. The authors can make it as part of future work or potential applications. Here, the zebra fish in the acetic environment showed higher frequency that the control. Higher tail beat frequency is directly related to higher velocities and power consumption based on previous work. Therefore, CFD was not really required to show this. This aspect needs to be softened in the manuscript.
Specific comments:
Equations 1 and 2 are not equations. There are no = sign. Please correct.
Equation 7b: sigma_j is not defined. It might be a typo.

·

Basic reporting

Please see the general comment section below.

Experimental design

The research question is well defined and the aim is clearly stated. The investigation performed meets the technical and ethical standard.

Methods are described well with the relevant references provided in the method section.

Validity of the findings

The CFD model has been validated in two different fish model with differing morphology. The authors have tested the CFD model in both the control and treated conditions. However, this study would have been robust if the authors perform experiment for different treatment conditions. Please, look at my general comments below.
The discussion of the results is limited mostly to supporting results with the authors mentioning the limitations of this model as well.

Additional comments

Review comments:
In this manuscript titled QUANTIFICATION OF THE INFLUENCE OF MEDICINE ON ZEBRAFISH LARVAE SWIMMING KINEMATICS AND ENERGETICS, the authors have tried to develop a zebrafish CFD model for objectively evaluating the effect of drugs in muscle power and energy consumption. The authors have described clearly their objectives and discussed the limitations of their study. Also, the model has been validated in the jellyfish model which has completely different morphology compared to the zebrafish larvae. However, there are some major and minor points that the authors can do to give clarity to the manuscript.
Major points:
1) Major drawback of this study:
In line 499-500 of discussion, the authors have speculated that this approach may be able to evaluate the potential analgesic drugs for pain relief. While potentially true, this study uses only one drug to develop a model. It is better if the authors try to test this zebrafish larvae- CFD model in at least 3 drugs (positive effect, no effect and negative effect) with diverse effects.
As acetic acid increases the movement in zebrafish larvae, could this model also be used to study the drugs that decreases the locomotion in zebrafish larvae? The authors have focused on the benefit of this model in developing drugs affecting nociception. But can this model have broader implications, not only in the analgesic drugs, but also in neuroactive drugs?
2) The authors should be more descriptive in the figures. For example, please see the following comments regarding the figures.
Figure 1: In line 135, 500 fps is mentioned, while in the figure, there is 300 fps.
Figure 3:
In the corresponding results section in the text of the manuscript, authors have not mentioned about the acetic acid treatment.
Also, the authors need to make this figure clear. It would be good for the readers if authors mention the full form of M, MIT and F also in the figure. So as the unit of force-mN.
Also, its difficult to understand if this is the result done in simulation zebrafish larvae model or in zebrafish larvae.
Figure 4: there is a lack of clarity in this figure. Authors have to mention what are D max, theta 1, theta 2, theta 3.
Figure 7: Authors have to be more descriptive about this figure as it is difficult to understand what they are trying to explain. For example: they have to mention what the vortex rings in the right side and left side of the panel represent. Also, it would be good to mention to the readers what to look into when they are looking at the vortex rings on the right and left panels.
Figure 8 b: Bars at 1 and 10 are incomplete. Also, authors have mentined this bar as a comparison between hydrodynamic power and mechanical power. But I only see the hydrodynamic power in the figure.
Figure 10: It is unclear, specifically the scale bar showing the grade of Z vorticity. Also, there is no labelling of the figures in both the left and right panel (ditto as figure 7). It would be better if the authors describe the differences between the figures in right and left panel by placing arrow mark/arrow head/ circle. It would help the readers understand the results that the authors have described in the corresponding text in the manuscript (line 438-440).
3) Line 85: I would request authors to change medicines to drugs, not only here, but also throughout the manuscript.
4) Line 105: I think pharmacological study would be better rather than medication study.


Minor points:
Line 82-83: The authors have written that “pure observation alone is not effective in ascertaining the influence of drugs in swimming behavior of fish”. If I understood correctly, the authors are referring this sentence to line 73-74. However, this is not only pure observation, its rather an objective quantification of the zebrafish larvae behavior based on distance travelled and velocity.
Line 83-84: This line is unclear. “Fish swimming kinematics are completely controlled by internal muscle and body” Authors have to clarify this sentence specifically the role of central nervous system and peripheral nervous system in swimming.
Line 117 -118: Authors have to clarify/restructure the phrase “how this study has established a foundation for the development of new analgesic drugs”.
Line 129-131: The total number of embryos per group is unclear. 20 or 40 embryos for each group should be clearly mentioned.
Line 344: “gestures of CFD model” should be clarified or rewritten.
Line 373-374: Authors have to restructure the sentence. It is not scientifically sound to mention that “Energy is passed along the fish body”.
Line 375-376: Author has to make this sentence clear, specifically the “Passive viscous dissipation of fish body”. It would be better to give an estimated energy (for example: 5% or 1%) rather than writing some energy.
516: The phrase “mystery of body stiffness” should be written scientifically.
518: Rather than “influences on muscle tissue”, effect on muscle tissue is appropriate.
The manuscript will be better with thorough editing and use of scientific terms.
Suggestions to the authors:
In this study, the authors have taken into account the movement of zebrafish larvae in the light. While zebrafish larvae move more in the dark and less in the light, it would be interesting to see the corresponding results in the dark period.

---

## Round 0.2 · Minor Revisions

Thank you for addressing the reviewer concerns. As there was a section of new work based specifically on a reviewer's suggestions, it was sent back to them for consideration. While they thought this addressed all their concerns, they did bring up a few grammatical/language-related issues that I want to give you the opportunity to address (or choose to leave the way you have written it). These are all minor edits and so such revisions to the manuscript will not lead to a re-review. We would presumably move to "accept" following the inclusion of any language-related edits you wish to make.

·

Basic reporting

no comment

Experimental design

no comment

Validity of the findings

no comment

Additional comments

The authors have addressed the issues I have raised that were within the scope of their study. Now, the manuscript is looking good. To further make it better, I expect the authors to thoroughly proofread the paper to correct the minor grammatical and spelling errors For example: I noticed that the authors have written "wildly" in line 62, which I think they mean widely. Similarly, in line 579, the authors have stated that "our future work could focus on evaluating potential analgesic drugs". I would prefer to say "would" rather than could. Please, look at this type of mistakes and correct them. It would be beneficial for the authors as well as the readers.
Other minor comments:
Line 431: Please replace the word deform with bend. Deform indicates abnormal morphology.
Line 568: In this line, it would be better to write replicate if the authors mean the results of this study are similar to the previously published studies.
Line 563-565: please provide clarity to the phrases "development stages" and "reversed results".

---

## Round 0.3 · accepted · Accept

Thank you for addressing the reviewer concerns and congratulations again.